# MODEL-BASED TRANSFER RL WITH TASK-AGNOSTIC OFFLINE PRETRAINING

## ABSTRACT

Pretraining RL models on offline datasets is a promising way to improve their training efficiency in online tasks, but challenging due to the inherent mismatch in dynamics and behaviors across tasks or data domains. We present Vid2Act, a model-based RL method that learns to transfer potentially useful dynamics and action demonstrations from various offline datasets to a novel task. The main idea is to use the world models not only as simulators for behavior learning but also as tools to measure the task relevance for both dynamics representation transfer and policy transfer. Specifically, we build a time-varying, task-selective distillation loss to generate a set of offline-to-online similarity weights. These weights serve two purposes: (i) adaptively transferring the task-agnostic knowledge of physical dynamics to facilitate world model training, and (ii) learning to replay relevant source actions to guide the target policy. We demonstrate the advantages of Vid2Act over the state-of-the-art methods in Meta-World and DeepMind Control Suite.

## 1 INTRODUCTION

Reinforcement learning (RL) approaches have made significant advancements in solving a wide range of sequential control problems (Ebert et al., 2018; Sekar et al., 2020; Laskin et al., 2020b). In the realm of visual RL, agents need to not only conduct representation learning from raw image inputs but also perform behavior learning in the learned state space, which requires a large number of interactions with an online environment and limits the applications in the real world. Recently, model-based RL algorithms have greatly improved sample efficiency by concurrently learning a differentiable simulator of the environment (*i.e.*, the world model), and using imagined rollouts generated by the world model for policy optimization (Kaiser et al., 2019; Hafner et al., 2020). Nevertheless, the process of training an effective world model from scratch remains a time-consuming and challenging pursuit, often yielding less generalizable representations.

To address this problem, many recent approaches (Stooke et al., 2021; Sun et al., 2023; Xu et al., 2023; Taiga et al., 2023) adopt the *pretraining and finetuning* paradigm to pre-learn representation models on off-the-shelf offline datasets and transfer the learned prior knowledge to a novel online RL domain. For example, SMART (Sun et al., 2023) exploits a Transformer model to learn generalizable visual representations from reward-free, offline interaction data under a control-centric pretraining objective. Similarly, our focus lies in leveraging multi-task offline data without reward to improve the visual RL performance in a novel online task. However, it is crucial to recognize that, despite the effectiveness of the pretraining method, a straightforward finetuning method may still suffer from the potential discrepancy in visual observations, physical dynamics, or even action spaces across task domains. Therefore, unlike SMART, our approach aims to:

1. *Adaptively identify the relevance between offline and online tasks in an unsupervised manner, allowing for positive domain transfer even when some offline data may seem unrelated.*

2. *Exploit relevant actions from the offline datasets to guide policy optimization for the new task.*

As shown in Figure 1, we propose a new *task-selective* transfer RL approach called Vid2Act to reduce the potential discrepancies between the pretraining stage and the transferring stage. In the pretraining stage, we exploit reward-free offline trajectories of image-action pairs to train task-specific world models, which learn the observation-to-state mapping functions and state-to-state transition functions for different source tasks. In the transferring stage, instead of performing direct finetuning,

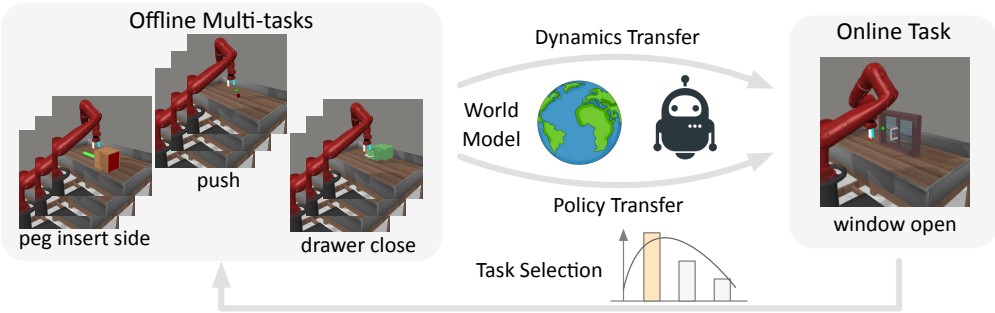

Figure 1: We aim to build an offline-to-online transfer RL agent for visual control problems, which is challenging due to the discrepancies between the target task and the source tasks from which the offline datasets are collected. The key idea of our approach is to leverage the world models to enable positive knowledge transfer through *task-selective dynamics distillation* and *behavior guidance*.

we leverage the source world models as teacher models to provide flexible regularization to the representation learning process of the target-domain agent. This is achieved through a *task-selective distillation loss*, where we learn a set of importance weights over the teacher models to adaptively transfer the prior knowledge of physical dynamics gathered from the offline data to the target world model.

In addition to their impact on representation learning, the importance weights also directly contribute to the policy optimization process conducted over the imaginations of the target world model. Specifically, Vid2Act incorporates a "*generative action replay*" module. During behavior learning, it serves to reproduce source-domain actions based on the target-domain states, which have been aligned to the corresponding source-domain state spaces by the distillation loss. By reusing the importance weights, we can dynamically select the most relevant source task at different time steps and replay its source expert behaviors to provide effective guidance for target policy improvement. We claim the proposed method to be "*task-agnostic*" because our experimental results show that it consistently enables Vid2Act to achieve positive domain transfer even when the available source offline data appears less relevant to the target task.

We evaluate Vid2Act on the Meta-World benchmark (Yu et al., 2020) and the DeepMind Control Suite (Tassa et al., 2018). Our approach shows remarkable performance improvements over both the vanilla model-based RL baselines like DreamerV2 (Hafner et al., 2021) and existing unsupervised pretraining methods for transfer RL, such as APV (Seo et al., 2022) and SMART (Sun et al., 2023).

## 2 RELATED WORK

**Visual RL.** In visual control tasks, the agent needs to learn policy from high-dimensional and complex observations. Learning generalized representation by either unsupervised (Gelada et al., 2019; Laskin et al., 2020a; Zhang et al., 2020; Schwarzer et al., 2020; Stooke et al., 2021; Yarats et al., 2021) or self-supervised manners (Choudhary et al., 2022; Yang et al., 2022; Ze et al., 2023), is a natural way to learn an auxiliary encoder of images for visual control tasks. Prior approaches consist of model-based methods to optimize latent dynamics model (Hafner et al., 2020; 2019; 2021; Pan et al., 2022), and model-free methods to utilize data augmentation (Laskin et al., 2020b; Yarats et al., 2020; Hansen et al., 2021; Cho et al., 2022) and contrastive representation learning (Oord et al., 2018; Anand et al., 2019; Mazoure et al., 2020; Laskin et al., 2020a; Nair et al., 2022; Li et al., 2023). Similar to our work, there are several methods that pretrain RL models on offline datasets and then finetune them on the online target task (Finn et al., 2016; Dwibedi et al., 2018; Zhan et al., 2020; Laskin et al., 2020a; Stooke et al., 2021; Schwarzer et al., 2021; Seo et al., 2022). Except for not bridging the domain gap between pretraining source data and RL tasks, they have shown attractive performance on vision-based RL tasks. In our framework, we do not directly finetune the parameters of the pretrained models, but rather learn more useful world models by distillation.

**Transfer RL.** Previous experiences across a diverse range of tasks can be beneficial in solving online control tasks, even when encountering them for the first time. To quickly leverage the past information to the new environments, many transfer learning approaches (Ng et al., 1999; Wiewiora et al., 2003; Li et al., 2014; Hinton et al., 2015; Rusu et al., 2015; Long et al., 2015; Gupta et al., 2016; Rebuffi et al., 2017; Yin & Pan, 2017; Zhang & Ma, 2018; Hester et al., 2018; Nair et al.,

2018; Marom & Rosman, 2018; Rebuffi et al., 2018; Xuhong et al., 2018; Liu et al., 2019; Li et al., 2019; Yao et al., 2020; Ma et al., 2019; Klissarov & Precup, 2020; Yang & Nachum, 2021; Zheng et al., 2022; Kadokawa et al., 2023) are proposed to bridge the gap across different tasks or domains. APV (Seo et al., 2022) employs action-free videos of multiple domains to pretrain an action-free recurrent state-space model (RSSM), which focuses on learning visual representation from offline datasets. Xu et al. (2023) trains a world model by using multiple offline tasks both in pretraining and finetuning to overcome the challenges of catastrophical forgetting. Recently, some methods leveraging Transformer have been proposed to facilitate transfer learning in control tasks (Xie et al., 2023; Sun et al., 2023). SMART(Sun et al., 2023) designs a control-centric pretraining objective for Decision Transformers(Chen et al., 2021) to captures the common essential information relevant to short-term control and long-term control across tasks. A work closely related to our approach is Knowledge Flow(Liu et al., 2019), which involves training multiple teacher models and distilling knowledge from their layers to a student model. In our work, we propose a task-selective distillation strategy to fully utilize both the dynamics and action information from the source tasks. It introduces a more flexible way to adaptively transfer useful knowledge to help downstream tasks.

## 3 PROBLEM FORMULATION

In the visual control task, the agent learns the behavior policy directly from high-dimensional observations, which is formulated as a partially observable Markov decision process (POMDP) with a tuple $(\mathcal{O}, \mathcal{A}, \mathcal{T}, \mathcal{R})$. Here, $\mathcal{O}$ is the observation space, $\mathcal{A}$ is the action space, $\mathcal{R}(s_t, a_t)$ is the reward function, and $\mathcal{T}(s_{t+1} \mid s_t, a_t)$ is the state-transition distribution. At each timestep $t \in [1; T]$, the agent takes an action $a_t \in A$ to interact with the environment and receives a reward $r_t = \mathcal{R}(s_t, a_t)$. The objective is to learn a policy that maximizes the expected cumulative reward $\mathbb{E}_p[\sum_{\tau=1}^{T} r_\tau]$.

To improve policy learning and sample efficiency of visual RL, we aim to transfer previous knowledge from multiple offline tasks. The offline datasets are reward-free and exclusively consist of image-action pairs $\{(o_t, a_t)\}$. It is important to note that there might be substantial distribution shifts in observations ($\mathcal{O}$), state transition functions ($\mathcal{T}$), and behaviors ($\mathcal{A}$) across task domains, which pose significant challenges in transfer learning, providing strong motivation for the development of a dynamic task-selective transfer RL approach. The primary goal of our approach is to efficiently bridge the gap between tasks in terms of state representations, physical dynamics, and action behaviors.

## 4 METHOD

In this section, we present a comprehensive overview of the pretraining process in the source datasets and the subsequent transfer learning process in the target task. The transfer learning process consists of two stages, *i.e.*, *task-selective dynamics transfer* and *behavior learning with generative action replay*, as shown in Figure 2 and described in detail in Algorithm 1.

### 4.1 PRELIMINARY: WHY MODEL-BASED RL FOR DOMAIN TRANSFER?

Our overall pipeline is built upon model-based RL, which involves learning the underlying dynamics from a buffer of past experiences, optimizing the control policy through future rollouts of compact model states, and executing actions in the environment to append the experience buffer. More precisely, we introduce a transfer RL approach based on the model-based DreamerV2 method (Hafner et al., 2021). Unlike previous work, the world model in our approach serves not only as a simulator for policy learning but also provides a measure of task relevance for both dynamics representation transfer and behavior transfer discussed in the following sections. Additionally, after pretraining the world models, subsequent algorithms can rely on the fixed parameters of the source models, making it more universal in real-world scenarios and decoupled from the source data.

### 4.2 MULTI-TASK OFFLINE PRETRAINING OF WORLD MODELS

As illustrated in Figure 2, we consider multiple reward-free, action-conditioned datasets denoted as $\mathcal{D}$. These datasets comprise expert data that has been previously collected from $N$ tasks and is readily available for our use. Our initial step involves pretraining a batch of action-conditioned video prediction models with independent model parameters, denoted as $\{F_\phi^i\}_{i=1}^N$. In contrast to APV (Seo et al., 2022), an existing model-based pretraining-finetuning transfer RL method, our approach incorporates actions during the pretraining phase, which is reasonable in learning the consequences

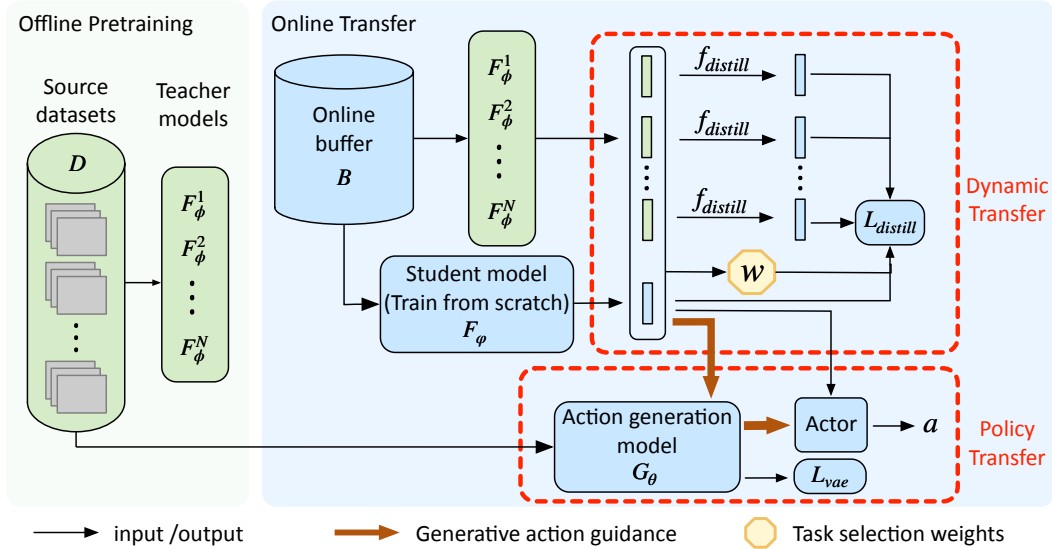

Figure 2: **Left:** We employ multiple offline action-conditioned video datasets to train multiple teacher models, whose parameters are frozen during the subsequent transfer learning process. **Right:** We dynamically distill prior knowledge from the teacher models for dynamics representation learning, and acquire a set of task-similarity weights $\mathcal{W}$. These weights are further used to reproduce the most relevant source actions to guide the target policy through a generative action replay model $G_\theta$.

of state transitions. Similar to Dreamer (Hafner et al., 2020), the pretrained models consist of three main components as follows:

$$
\begin{aligned}
\text{Representation model:} \quad & q(s_t \mid s_{t-1}, a_{t-1}, o_t) \\
\text{Dynamics model:} \quad & p(\hat{s}_t \mid s_{t-1}, a_{t-1}) \\
\text{Decoder model:} \quad & p(\hat{o}_t \mid s_t).
\end{aligned}
\tag{1}
$$

The representation model extracts posterior latent states $s_t$ from observations $o_t$, previous states $s_{t-1}$, and previous actions $a_{t-1}$. The dynamics model follows the Recurrent State Space Model (RSSM) architecture from PlaNet (Hafner et al., 2019) to predict the prior latent states $\hat{s}_t$ without access to the corresponding $o_t$. The decoder reconstructs $\hat{o}_t$ given the latent states. All components are optimized jointly using the following loss function:

$$
\mathcal{L}_{\text{source}} = \mathbb{E} \left\{ \sum_{t=1}^{T} \underbrace{-\ln p(o_t \mid s_t)}_{\text{Image reconstruction}} + \underbrace{\beta \, \text{KL}[q(s_t \mid s_{t-1}, a_{t-1}, o_t) \parallel p(s_t \mid s_{t-1}, a_{t-1})]}_{\text{KL divergence}} \right\},
\tag{2}
$$

where $\beta$ is a hyperparameter of the Kullback-Leibler (KL) divergence that regularizes the approximate posterior learned from the representation model toward the prior learned from the dynamics model.

### 4.3 TASK-SELECTIVE DYNAMICS TRANSFER

It is important to note that, even though the pretraining method is effective, a simple finetuning approach may encounter challenges due to the potential discrepancy in visual observations, physical dynamics, or even action spaces across task domains. Therefore, when a novel target task emerges, we initialize a student world model $F_\varphi$ from scratch, while freezing the parameters of $\{F_\phi^i\}_{i=1}^N$ and using them as teacher models to transfer the dynamics representations from the source domains (see Figure 2). In addition to the model components outlined in Equation (1), $F_\varphi$ also incorporates a reward model represented as $\hat{r}_t \sim p(r_t \mid s_t)$. To avoid confusion of notations, we use $s_t^i$ to denote the state of $i$-th pretrained teacher model, and $e_t$ to denote the state of the target student model. Given a latent state denoted by $e_{t-1}$ and a corresponding action $a_{t-1}$, we first transit this state to the next time step individually using the teacher models and the student model, obtaining $\{s_t^i \sim p(e_{t-1}, a_{t-1}; \phi)\}_{i=1}^N$ and $e_t \sim p(e_{t-1}, a_{t-1}; \varphi)$.

To close the distance between the marginal distributions of state transitions produced by the source world model and the dynamics estimated by each teacher model, we incorporate a distillation network

---

**Algorithm 1:** Vid2Act with improved dynamics learning, behavior learning & policy deployment

---

1  **Hyperparameters:** $H$: Imagination horizon
2  Initialize the offline replay buffer $\mathcal{D}$ with multiple tasks.
3  Initialize the online replay buffer $\mathcal{B}$ with random episodes.
4  **while** *not converged* **do**
5    **for** *update step* $c = 1 \ldots C$ **do**
6       Draw data sequences $\{(o_t, a_t, r_t)\}_{t=1}^{T} \sim \mathcal{B}$.
7       `// Dynamics learning and source action replay`
8       Compute distillation loss using Equation (4) and update world model parameters using Equation (5)
9       Draw data sequences $\{(o_t, a_t)\}_{t=1}^{T} \sim \mathcal{D}$ from each source task.
10      Compute action generation model loss using Equation (7) and update model parameters.
11      `// Behavior learning`
12      **for** *time step* $i = t \ldots t + H$ **do**
13        Select the task label $k$ with highest confidence according to Equation (3)
14        Imagine an action $a_i \sim \pi(a_i \mid e_i, G_\theta(e_i, k))$.
15        Predict rewards $r_i \sim p(r_i \mid e_i)$ and values $v_\psi(e_i)$
16      **end**
17      Update the policy and value models in Equation (8) using estimated rewards and values.
18    **end**
19    `// Environment interaction`
20    $o_1 \leftarrow$ `env.reset()`
21    **for** *time step* $t = 1 \ldots T$ **do**
22      Calculate the posterior representation $e_t \sim q(e_t \mid e_{t-1}, a_{t-1}, o_t)$ from history.
23      Use the teacher models to obtain $\{\hat{s}_t^i \sim p(e_{t-1}, a_{t-1}; \phi) \mid i \in [1, N]\}$ and determine the task label $k$ with highest confidence according to Equation (3)
24      Compute $a_i \sim \pi(a_i \mid e_i, G_\theta(e_i, k))$.
25      $r_t, o_{t+1} \leftarrow$ `env.step`$(a_t)$
26    **end**
27    Add experience to the online replay buffer $\mathcal{B} \leftarrow \mathcal{B} \cup \{(o_t, a_t, r_t)_{t=1}^{T}\}$.
28  **end**

---

in $F_\varphi$, denoted as $f_{\text{distill}}$, which takes the form of a multilayer perceptron (MLP). The role of this module is to extract transferable features from the predicted states of the teacher models. In other words, it transforms the states $s_t^i$ predicted by the teacher models into a set of transferable features $\{u_t^i = f_{\text{distill}}(s_t^i)\}_{i=1}^{N}$. These features are then used in the knowledge distillation loss.

Intuitively, each source task may hold varying impacts on the dynamics learning of the target visual control task. We introduce the concept of task-similarity weights and propose to optimize these weights through the knowledge distillation loss. By learning this set of weights, we can dynamically transfer knowledge in an adaptive manner based on offline-online task relevance. To compute the similarity weight $\mathcal{W}$, we concatenate the predicted state $s_t^i$ of each teacher model and the predicted state $e_t$ of the student model. This concatenated representation is then fed into a fully-connected layer $f_{\text{weight}}$, followed by a softmax activation function:

$$\text{Task selection weights:} \quad \mathcal{W} = \{w_i\}_{i=1}^{N} = \text{Softmax}(\{f_{\text{weight}}(s_t^i * e_t)\}_{i=1}^{N}), \tag{3}$$

where $*$ denotes the operation of concatenation. We then minimize the Euclidean distance between pairs of states as follows, taking into account the corresponding task-similarity weights:

$$\mathcal{L}_{\text{distill}} = \sum_{i=1}^{N} \sum_{t=1}^{T} w_i \cdot \parallel e_t - u_t^i \parallel_2^2. \tag{4}$$

The overall objective of the student model can be written as follows, where $\alpha$ is a hyperparameter:

$$\mathcal{L}_{\text{target}} = \mathbb{E}\left[\left[\sum_{t=1}^{T} \underbrace{\beta \, \text{KL}\big[q(e_t \mid e_{t-1}, a_{t-1}, o_t) \parallel p(e_t \mid e_{t-1}, a_{t-1})\big]}_{\text{KL divergence}} \right.\right.$$
$$\left.\left. \underbrace{- \ln p(o_t \mid e_t)}_{\text{Image reconstruction}} \underbrace{- \ln p(r_t \mid e_t)}_{\text{Reward prediction}} \right] + \alpha \, \mathcal{L}_{\text{distill}}\right]. \tag{5}$$

Equation (4) is the fundamental basis for Vid2Act. When the dynamics of the source domain are similar to the target task, the latter term of this loss naturally becomes smaller. On the other hand, for source tasks with significantly different dynamics from the target task, the model will minimize the weight term in order to minimize this loss. The use of the *task-selective* distillation loss allows the student model to effectively learn from multiple teachers while considering their respective importance weights. This approach grants the student model the ability to acquire significant prior knowledge regarding intricate physical dynamics from the most relevant source tasks. By selectively distilling knowledge from these source tasks, the student model can adapt and incorporate valuable information to enhance its overall learning capabilities.

### 4.4 TASK-SELECTIVE BEHAVIOR TRANSFER

**Generative replay of selected source actions.**    In the representation learning stage for the target task, we reuse the offline source datasets to learn an action generation model with the explicit task label $k \in \{1, \ldots, N\}$ to provide guidance for target behavior learning. Inspired by BCQ (Fujimoto et al., 2019), which is an offline RL method, we design $G_\theta$ using a state-conditioned variational auto-encoder (VAE) (Kingma & Welling, 2013; Sohn et al., 2015). The action generation model $G_\theta$ consists of an encoder $E_{\theta 1}$ and a decoder $D_{\theta 2}$. The encoder takes a state-action pair and a task label $k$, and outputs a Gaussian distribution $\mathcal{N}(\mu, \sigma)$. The state $s$ of a source task distilled by $f_{\text{distill}}$, along with a latent vector $z$ sampled from the Gaussian distribution and a task label $k$, is passed to the decoder $D_{\theta 2}(s, z)$ which outputs an action:

$$\mu, \sigma = \mathrm{E}_{\theta 1}(s, a, k), \quad \hat{a} = \mathrm{D}_{\theta 2}(s, z, k), \quad z \sim \mathcal{N}(\mu, \sigma). \tag{6}$$

The action generation model $G_\theta$ is optimized by

$$\mathcal{L}_{\text{replay}} = \mathbb{E} \left[ \sum_{t=1}^{T} (a - \hat{a})^2 + \mathrm{KL}\left(\mathcal{N}(\mu, \sigma) \parallel \mathcal{N}(0, 1)\right) \right]. \tag{7}$$

**Behavior learning with generative action guidance**    We utilize an actor-critic algorithm to learn the policy over the predicted future state and reward trajectories. As shown in Figure 2, we use the action generation model $G_\theta$ to promote policy learning, which *1) provides an efficient indication when a strong correlation exists between the source and target tasks, and 2) expends exploration of action space when there is little correlation between them.* The parameters in $G_\theta$ are frozen at this stage. Notably, instead of using the output of $G_\theta$, we use the feature of $G_\theta$ before output, which is more informative. Reusing the similarity weights learned in dynamics transfer, we can dynamically select task label $k$ with the highest confidence to generate action guidance. Therefore, we modify the action model and the value model as follows:

$$\text{Action model:} \quad a_t \sim \pi(a_t \mid e_t, G_\theta(e_t, k)), \quad \text{Value model:} \quad v_\psi(e_t) \approx \mathbb{E}_{\pi(\cdot \mid e_t)} \sum_{k=t}^{t+H} \gamma^{k-t} r_k, \tag{8}$$

where $H$ is the imagination time horizon. Similar to the process of behavior learning, we also utilize the action generation model $G_\theta$ to draw action from the action model during policy deployment. As shown in Lines 23-24 in Algorithm 1, the action guidance is dependent on current states $e_i$ and the source task label with the highest task-similarity weights, which may evolve over time.

## 5 EXPERIMENTS

### 5.1 EXPERIMENTAL SETUP

We evaluate Vid2Act on two visual RL environments in an offline-to-online domain transfer setup:

- **Meta-World** (Yu et al., 2020): It simulates 50 manipulation tasks, all involving the same robotic arm. We collect 6 offline datasets using expert experiences from *button press topdown*, *door open*, *drawer close*, *peg insert side*, *pick place*, and *push*. Each of them contains 10 demonstrations.

- **DeepMind Control Suite** (Tassa et al., 2018): It is a standard benchmark for visual-based RL that contains a diverse set of continuous control tasks. We collect offline datasets from 4 tasks, *i.e.*, *cheetah run*, *hopper stand*, *walker walk*, and *walker run*. Each task contains 50 trajectories of expert experiences.

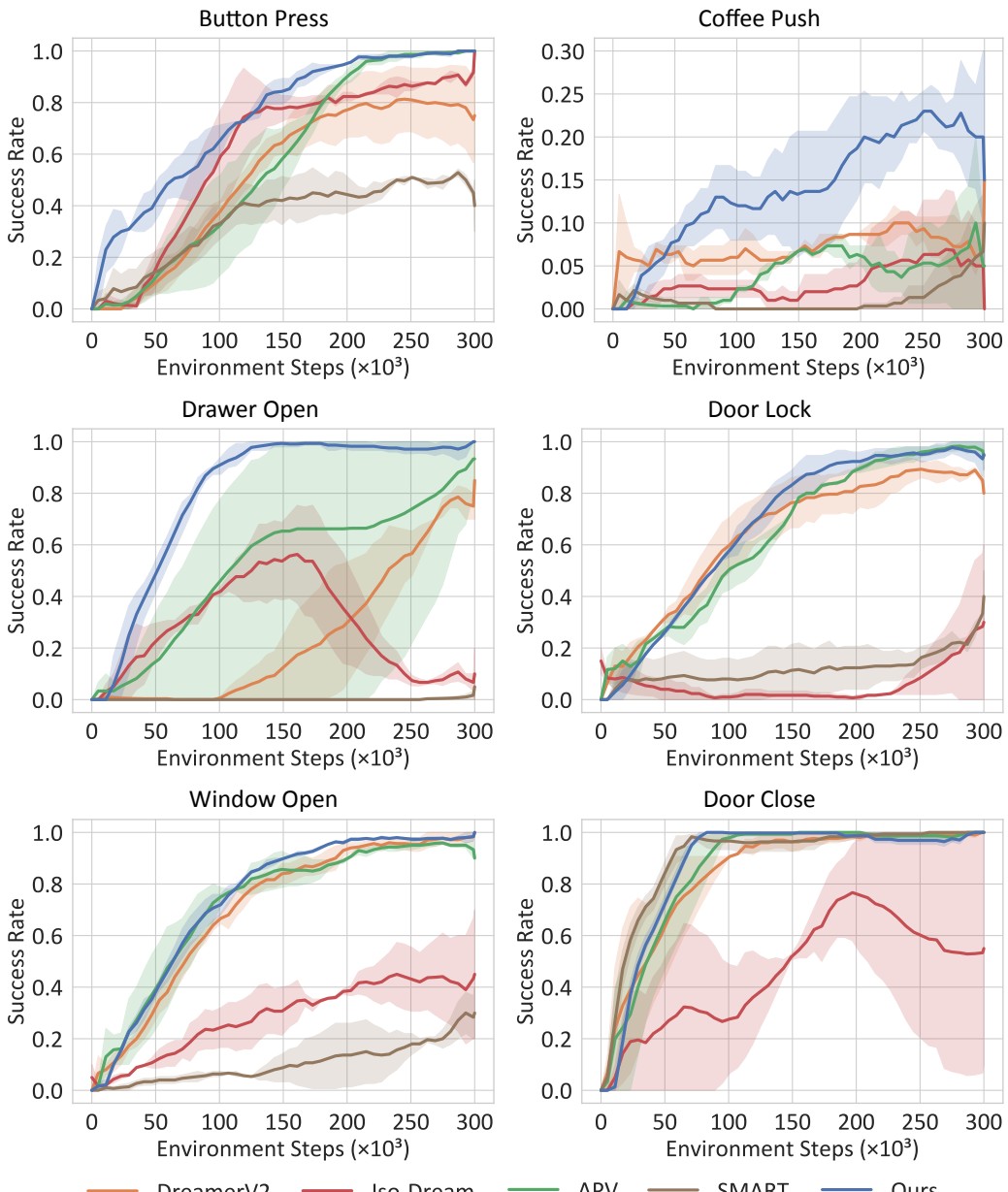

Figure 3: Performance comparison with the state-of-the-art methods on 6 tasks from Meta-World as measured on the success rate. Vid2Act generally outperforms the compared models.

We compare Vid2Act with the following approaches:

- **DreamerV2** (Hafner et al., 2021): A model-based RL method that learns the policy directly from latent states in the world model. For a fair comparison, we pretrain this model with offline datasets.

- **APV** (Seo et al., 2022): A model-based RL method that stacks an action-conditional RSSM model on top of the pretrained action-free RSSM model. It utilizes the pretrained representations for finetuning. We train this model by following its training setting.

- **Iso-Dream** (Pan et al., 2022): A strong baseline for visual RL method that focuses on modeling different dynamics based on controllability. It rolls out noncontrollable states into the future and performs policy optimization based on the decoupled latent imaginations.

- **SMART** (Sun et al., 2023): A generic multi-task pretraining framework that designs a Control Transformer coupled with a control-centric pretraining objective in a self-supervised manner.

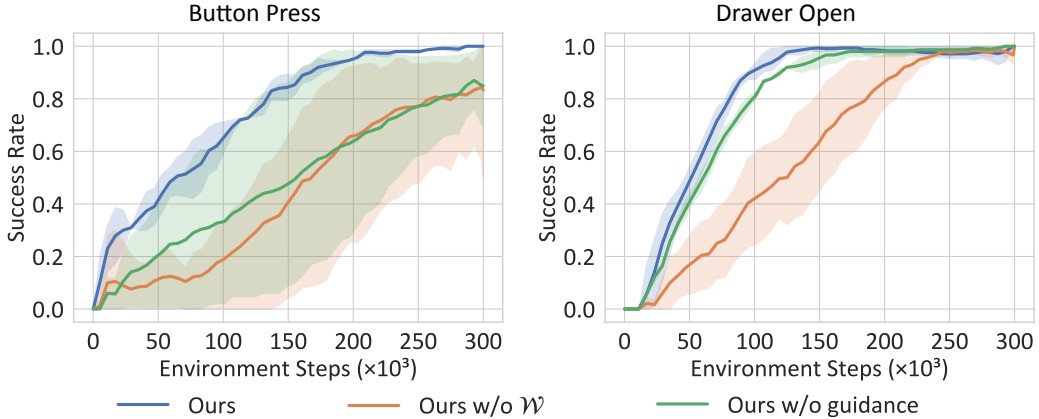

Figure 4: Ablation studies of Vid2Act that shows the impact of learning time-varying task importance weights (orange) and optimizing behavior learning with generative action guidance (green).

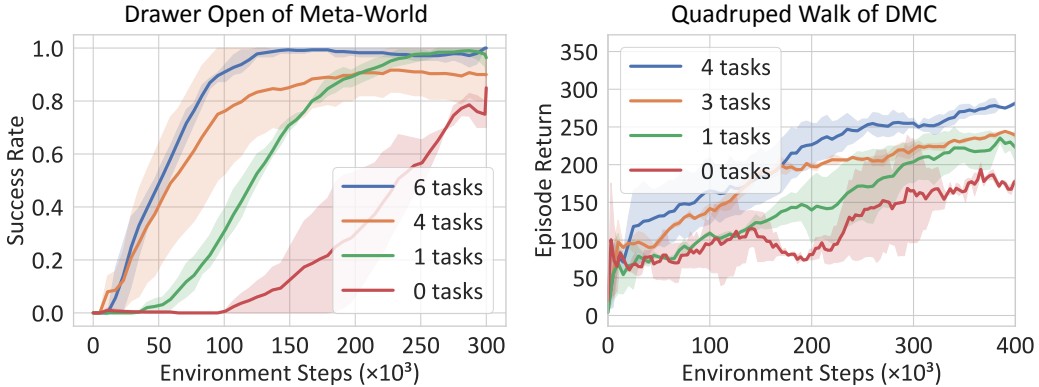

Figure 5: Analyses on the impact of different source task configurations.

## 5.2 META-WORLD

**Implementation details.** We first pretrain the teacher models of action-conditioned video prediction model by minimizing the objective in Equation (2) for $200K$ gradient steps. Our model is evaluated in 6 tasks, *i.e.*, *button press*, *coffee push*, *drawer open*, *door lock*, *window open*, and *door close*. In all tasks, the episode length is 500 steps without any action repeat. The number of environment steps is limited to $300K$. We run all tasks with 3 seeds and report the mean success rate and standard deviations of 10 episodes.

**Quantitative comparisons.** As shown in Figure 3, our Vid2Act generally outperforms other methods in 6 tasks. We improve DreamerV2 about $25\%$ in *button press* and $90\%$ in *coffee push*. Iso-Dream, which serves as a robust baseline for addressing visual control tasks using isolated state transition branches, exhibits limitations in handling a majority of the tasks in Meta-World. Compared with APV, our model is more stable and effective, especially in *drawer open*.

**Ablation studies.** We conduct ablation studies to confirm the validity of learning a set of time-varying task importance weights and behavior learning with generative action guidance on two tasks, as shown in Figure 4. Without the process of learning the importance weights (orange) to measure the similarity between source and target tasks, the performance of our model has decreased by about $25\%$ in *button press*, and it requires more timesteps to improve the behavior policy in *drawer open*. It demonstrates that information in different source tasks has different impacts on the target task, and a task-selective knowledge distillation loss with importance weights encourages the student model to adaptively find useful prior knowledge and transfer it to help the dynamics learning in downstream tasks. Moreover, we evaluate Vid2Act without generative action guidance for behavior learning (green). The result shows that our proposed task-selective behavior learning strategy can identify potentially valuable source actions and employ them as exemplar guidance for the target policy.

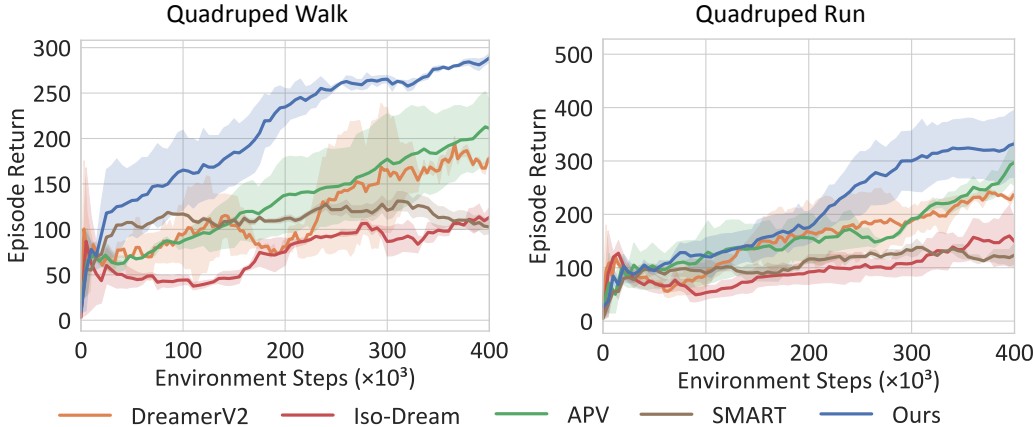

Figure 6: Performance comparison on two tasks from DeepMind Control Suite as measured on the episode rewards. Our Vid2Act with dynamic knowledge distillation achieves significant improvements compared with existing model-based RL approaches.

**Impact of fewer or less relevant source tasks.** To analyze the robustness of our approach to different source domain configurations, we sequentially decrease the number of source domain tasks according to the learned importance weights, *i.e.*, gradually removing the task with the highest importance weight. The results are shown in Figure 5. We have two observations in this figure. First, compared with the baseline model that is solely trained on the target task, our approach consistently achieves positive offline-to-online transfer even when it can only access parts of the source datasets with lower importance weights. Second, as the number of the source tasks grows, the performance of Vid2Act improves as well, demonstrating its effectiveness in identifying task similarity and improving the target policy with the expanded offline datasets.

## 5.3 DeepMind Control Suite

**Implementation details.** In this environment, the episode length is $1,000$ steps with the action repeat of 2, and the reward ranges from 0 to 1. We pretrain four teacher models for four source tasks with $200K$ gradient steps. For the online target tasks, we train our method for $200K$ iterations, which results in $400K$ environment steps. The input image size is set to $64 \times 64$, the batch size is 50, and the imagination horizon is 50. The hyperparameters $\beta$ and $\alpha$ are set to 1 in Equation (5).

**Quantitative comparisons.** We evaluate Vid2Act with baselines on the mean episode rewards and standard deviations. The results of *quadruped walk* and *quadruped run* are shown in Figure 6. Our framework achieves significant improvements compared with existing model-based RL approaches. For example, Vid2Act performs nearly 100 higher performance on the task of *quadruped walk* and *quadruped run* than DreamerV2 after $400k$ steps environment interactions. These results show that our model can effectively transfer valuable dynamics knowledge from source tasks. Compared with APV, which only uses the pretrained action-free world model as initialization to train downstream tasks, Vid2Act is encouraged to learn more precise state transitions based on action input and more useful source dynamics based on task-selective knowledge distillation. Moreover, the learned task selection weights help the agent adaptively transfer potentially useful action demonstrations from offline datasets. Such as *quadruped walk*, our agent tends to select a source task with the greatest relevance, *i.e.*, *walker walk* in this online task.

## 6 Conclusion

In this paper, we proposed a new task-selective transfer learning framework called Vid2Act that improves visual RL with offline datasets with multiple tasks. Vid2Act has two contributions. First, it provides a novel model-based pretraining and transfer learning pipeline for visual RL. Unlike APV (Seo et al., 2022), it transfers action-conditioned dynamics from multiple source tasks with a set of importance weights learned by the world models. Second, it provides a novel task-selective behavior learning strategy that identifies potentially valuable source actions and employs them as exemplar guidance for the target policy. Experiments in the Meta-World and the DeepMind Control environments demonstrated that Vid2Act significantly outperforms existing visual RL approaches.

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
