# OpenReview forum: "Model-Based Transfer RL with Task-Agnostic Offline Pretraining"
_ICLR.cc/2024/Conference — ICLR 2024 Conference Withdrawn Submission_

### Official Review · Reviewer_xfrk · 2023-10-23

**Soundness:** 2 fair
**Presentation:** 3 good
**Contribution:** 2 fair
**Rating:** 3
**Confidence:** 4

**Summary:**

Due to the inherent mismatch in dynamics and behaviors across tasks, effectively transferring pretrained RL models from diverse offline datasets is challenging. This paper proposes Vid2Act, which learns to transfer potentially useful dynamics and action demonstrations from various offline datasets to a novel task.  Vid2Act build a time-varying, task-selective distillation loss to generate a set of offline-to-online similarity weights to adaptively transfer the task-agnostic knowledge of physical dynamics and replay relevant source actions. Experiments on the Meta-World and DeepMind Control Suite demonstrates the advantage of Vid2Act.

**Strengths:**

1. This paper is well-written and easy to follow.
2. Transferring pretrained worlds models is an interesting and important topic for RL.

**Weaknesses:**

1. Assuming that there exist both an exact world model and expert demonstrations for each source task is too strong.
2. Empirical results shown in the paper may be not enough to illustrate the effectiveness of the proposed method. To be clear, in several tasks from the Meta-World (e.g., Window Open, Door Close, Door Lock), the success rate of Vid2Act converges to 1.0 without a remarkably faster speed compared with baseline methods. Moreover, there are only two tasks from DMC and in Quadruped Run, Vid2Act performs in par with APV after 400k steps (Figure 6).

**Questions:**

1. In section 4.4, you claimed that the feature of $G_\theta$ before output is more informative than the output of $G_\theta$, but why? And what do you mean by saying that "more informative"?
2. How do you verify that the learned weights indeed help select the most similar task?
3. The generative action guidance is trained on expert data from each source task, but tested on the training data from target task, will this distribution shift hurt?

---

### Official Review · Reviewer_E2SU · 2023-10-30

**Soundness:** 2 fair
**Presentation:** 1 poor
**Contribution:** 2 fair
**Rating:** 3
**Confidence:** 4

**Summary:**

In this paper it introduces Vid2Act, a model-based reinforcement learning framework that first pretraining of a collection of dynamics models utilizing a multitask offline dataset. Subsequently, it employs a state-dependent task similarity metric to selectively transfer the learned dynamics to pertinent downstream tasks. This targeted transfer mechanism ensures the applicability of dynamics solely from tasks that are relevant to the downstream tasks. Empirical evaluations performed on the DeepMind Control Suite and MetaWorld benchmarks demonstrate the effectiveness of the proposed approach.

**Strengths:**

The motivation of the proposed approach is clear, which is to only utilizing the prior knowledge of the environment dynamics that is only relevant to the downstream task.

**Weaknesses:**

The description of the proposed method overall is confusing at many places, and so the intuition of the proposed architecture is not very well justified. Please see my questions for details on this. In addition, the proposed algorithm also has the fundamental limitation that it has to learn a dynamics model per task. As a result, it becomes fundamentally impossible to scale up this approach to the scenarios where the number of pretraining tasks is large (Metaworld-45 or the more recent LIBERO benchmark).

The empirical evaluation is also problematic, with the choice of tasks being too simple (metaworld) or too few (deepmind control suite). More tasks and harder tasks, more environment interaction steps, more baselines, and more random seeds (although this is minor) are needed to demonstrate the effectiveness of the proposed approach.

Additionally, many relevant works are missing in the paper.

**Questions:**

### 1. Questions about task similarity weights calculation
**(a)** For calculating the task similarity score, why having a distillation layer \( f_\text{distill} \), why cannot we just directly take \( \|e_t - s_t^i\| \)? The role of \( f_\text{distill} \) and \( f_\text{weight} \) is very unclear even though the authors are trying to explain in the text.

**(b)** Why not a simpler approach such as [7]? We could calculate the task similarity by simply looking at the similarity of their gradients. This should serve as a baseline to substitute the proposed architecture here if the authors want to claim that learning \( f_\text{distill} \) and \( f_\text{weight} \) gives a better solution.

### 2. Clarification of Terms
What is \( G_\theta(e_t, k) \) in equation (8)? The author only mentions that \( G_\theta \) consists of the encoder \( E_{\theta_1} \) and decoder \( D_{\theta_2} \), and what \( E_{\theta_1} \) and \( D_{\theta_2} \) are in (6) and (7).

### 3. Handling Different Action Spaces
It is not clear to me how the proposed method handles different action space dimensionalities in DMC tasks.

### 4. Task Selection
- **(MetaWorld):** The six tasks from Metaworld are all too simple. The authors should consider more challenging tasks such as Hand Insert, Stick Pull, Stick Push, Disassembly, Assembly, Shelf Place, and so on. Please refer to [1] for a rough categorization of the difficulty of tasks in MetaWorld.
- **(DMC):** Only two tasks during downstream finetuning are considered (Quadruped Walk and Quadruped Run). More medium-difficulty tasks in DMC should be considered. Please refer to [3] for a list of medium-difficulty tasks. Additionally, a few harder humanoid tasks would be good to demonstrate the effectiveness of the proposed pretraining approach.

### 5. Number of Environment Steps
For DMC, 400K environment timesteps are far too few to draw a conclusion. With only 400K steps, the performance of all methods on Quadruped Run and Quadruped Walk is poor (<350 compared to 800-900 of the convergent performance). For Metaworld, the authors use 200K, but it is mostly that the selected tasks are too easy.

### 6. Baselines
**(a)** [1] and [8] are both model-based approaches for visual RL that should serve as baselines. Maybe [1] is more relevant since it’s dreamer-based, but at least [8] should be mentioned in the paper.
**(b)** Dreamer-v3 [2]: the author should compare with the latest dreamer model.
**(c)** Model-free approaches [3,4,5]: model-free visual RL algorithms should also be good baselines. ([3,4,5] are missing from the related works of the paper.) If after pretraining latent dynamics from multitask offline datasets, it still does not perform as well as the model-free visual RL approach from scratch, then the pretraining would be of no value.

### 7. More Random seeds
(Minor point): More random seeds are needed for the empirical evaluation. Right now Vid2Act only runs 3 random seeds.

### 8. Missing Related Works
[6] is a latent model-based transfer RL approach that should also be discussed in the related work section.


### References
- [1] Younggyo Seo, Danijar Hafner, Hao Liu, Fangchen Liu, Stephen James, Kimin Lee, Pieter Abbeel, "Masked World Models for Visual Control," CoRL 2022.
- [2] Danijar Hafner, Jurgis Pasukonis, Jimmy Ba, and Timothy Lillicrap, "Mastering Diverse Domains through World Models," Arxiv Preprint.
- [3] Denis Yarats, Rob Fergus, Alessandro Lazaric, Lerrel Pinto, "Mastering Visual Continuous Control: Improved Data-Augmented Reinforcement Learning," ICLR 2021.
- [4] Edoardo Cetin, Philip J. Ball, Steve Roberts, Oya Celiktutan, "Stabilizing Off-Policy Deep Reinforcement Learning from Pixels," ICML 2022.
- [5] Ruijie Zheng, Xiyao Wang, Yanchao Sun, Shuang Ma, Jieyu Zhao, Huazhe Xu, Hal Daumé III, Furong Huang, "TACO: Temporal Latent Action-Driven Contrastive Loss for Visual Reinforcement Learning," NeurIPS 2023.
- [6] Yanchao Sun, Ruijie Zheng, Xiyao Wang, Andrew E Cohen, Furong Huang, "Transfer RL across Observation Feature Spaces via Model-Based Regularization," ICLR 2022.
- [7] Yifan Xu, Nicklas Hansen, Zirui Wang, Yung-Chieh Chan, Hao Su, Zhuowen Tu, "On the Feasibility of Cross-Task Transfer with Model-Based Reinforcement Learning," ICLR 2023.
- [8] Nicklas A Hansen, Hao Su, Xiaolong Wang, "Temporal Difference Learning for Model Predictive Control," ICML 2022.

---

### Official Review · Reviewer_9i9x · 2023-11-01

**Soundness:** 3 good
**Presentation:** 3 good
**Contribution:** 2 fair
**Rating:** 5
**Confidence:** 3

**Summary:**

The paper provides an interesting method for the paradigm of offline to online transfer RL. The method introduces a way to use multiple offline trained models in a weighted manner to learn a new model for a new online task. They don't use the offline world models just as a simulator to roll out trajectories but are instead able to extract task-relevant dynamics and behaviour from offline models.

**Strengths:**

* Interesting student-teacher approach regarding finding task-similarity weights and using them towards the representation learning of online model.
* To my knowledge, also novel to use offline models towards both representation and behaviour learning.

**Weaknesses:**

* Experiments:
    - Weaker or equivalent performance compared to other MBRL methods that do not even use offline data.
       Compare Figure 3 from this paper to Figure 14 in TDMPC [1] and compare figure 6 from this paper to figure 4 in [1].
    - Not trained for long enough steps to compare the optimal policy for DM control tasks
* Assumed individual task world models. Will the method work with a single offline model trained with trajectories from multiple offline datasets?
* Assumes access to the offline dataset as well for training action generation module even though training offline models. If direct access to the offline dataset is needed, why not use the offline data directly as well for representation learning for the online task? I take the need for both offline data and offline model to be taken as an inefficiency of the method.
* Related to my 1st question, there is no quantitative or qualitative analysis of $w$ being learned and actual task similarity between the online model being learnt and offline models.

[1] Hansen, N., Wang, X., & Su, H. (2022). Temporal difference learning for model predictive control. arXiv preprint arXiv:2203.04955.

**Questions:**

* What's stopping the task-selective weight to be learned such that $w_i = 1, i=k \quad \text{and} \quad w_i = 0, i != k$  for some single $k$, where $k$ might be the offline model which is closest to the current online task. I.e The current task would only care about one of the offline models.
* Can the method also be worked with proprioceptive observations?

---

### Official Review · Reviewer_oHAr · 2023-11-02

**Soundness:** 2 fair
**Presentation:** 3 good
**Contribution:** 3 good
**Rating:** 5
**Confidence:** 4

**Summary:**

A key motivation for model-based reinforcement learning is the promise of generalization / transfer of knowledge to new tasks. This work considers the problem of transferring models trained on offline datasets of observation-action pairs (no rewards) for a number of source tasks to a new target task for which we have access to a reward function. The proposed framework, dubbed Vid2Act, trains task-specific world models for individual source tasks, and then leverage the learned models during the transfer stage in which a new model is learned from scratch on the new task using online RL. Specifically, the source models are used to distill knowledge into the new student model, weighted by their relevance to the target task. Additionally, Vid2Act also uses a novel *generative replay* where the behavior policy for the target task is additionally provided with the embedding of an action generation model trained on source data. Experiments are conducted on visual DMControl + Meta-World tasks in an offline-to-online setting, and the proposed method is consistently on par or better than (seemingly) strong baselines.

**Strengths:**

- Paper is well written and easy to follow. The problem setting is timely, well motivated, and likely to be relevant to the ICLR community. There is sufficient discussion of related work to contextualize the technical contributions.
- Experiments are conducted on standard benchmarks and the overall experimental design appears to be rigorous. The authors compare their proposed method against seemingly strong baselines.
- Ablations are useful for understanding the relative importance of each component.

**Weaknesses:**

- While the technical contributions are interesting and the proposed method empirically appears to be competitive, the paper currently lacks analysis and insights into *why* the method works well. The authors repeatedly note that the design of the distillation loss and task weighting permits transfer of knowledge across tasks even when the tasks are seemingly dissimilar (and the results in Figure 5 empirically show that this seems to be the case), but I feel that the paper could be a lot more insightful if it wasn't strictly focused on beating the baselines. For example, it would be helpful to visualize how the task weights change over the course of training, and whether the discovered task similarities match human intuition / are semantically meaningful.
- The paper lacks discussion of limitations. For example, what if source tasks are distinctly different? E.g. transferring the DMControl source models to Meta-World and vice-versa. Would there be any benefit of pretraining compared to simply training a world model from scratch on the target task? What would the task weights look like in this setting? What are other likely failure modes / limitations of the proposed method? Experiments are only conducted with offline datasets that consist purely of expert behavior. Would the proposed method work for offline datasets of mixed-quality behavior?

**Questions:**

Addressing my comments listed in the *weaknesses* section is most likely to change my opinion. However, I'd also like the authors to clarify the following:
- In the case of the DMControl experiments, the authors only consider target tasks using the Quadruped embodiment which is not included in the offline dataset. Why did the authors choose these particular tasks for transfer? What would the results look like for a transfer setting in which the target task is a seen embodiment but unseen task? Intuitively, I would expect transfer to be most effective in this setting, since the dynamics are exactly the same -- only the task (reward function) is new. The Meta-World experiments don't quite cover this setting since objects and thus dynamics differ even though it is the same robot.
- Code is provided as supplementary material. Are the authors committed to open-sourcing their code in case of acceptance?